# In Vitro Evaluation of the Cytotoxic Potential of Thiosemicarbazide Coordinating Compounds in Hepatocyte Cell Culture

**DOI:** 10.3390/biomedicines11020366

**Published:** 2023-01-26

**Authors:** Valeriana Pantea, Vitalie Cobzac, Olga Tagadiuc, Victor Palarie, Valentin Gudumac

**Affiliations:** 1Laboratory of Biochemistry, Nicolae Testemitanu State University of Medicine and Pharmacy, 2004 Chisinau, Moldova; 2The Laboratory of Tissue Engineering and Cell Cultures, Nicolae Testemitanu State University of Medicine and Pharmacy, 2004 Chisinau, Moldova; 3Biochemistry and Clinical Biochemistry Chair, Nicolae Testemitanu State University of Medicine and Pharmacy, 2004 Chisinau, Moldova

**Keywords:** hepatocyte cells, MTT, resazurin, thiosemicarbazide coordinating compounds, Wistar male rats, cytotoxicity assay

## Abstract

Cancer is a global medical problem and, despite research efforts in the field of tumor treatment, there is currently a shortage of specific anticancer drugs. Most anticancer drugs show significant side effects. The liver is the organ that has central functions in drug metabolism, being a major target of the harmful action of anticancer compounds. In this context, it is essential to evaluate the cytotoxic effects of potential anticancer substances. Therefore, hepatotoxicity and hepatocyte viability were determined in vitro to evaluate the action of seven new local thiosemicarbazide coordination compounds (CCT) on normal liver cells. Doxorubicin was used as a reference substance. The control group consisted of hepatocytes not exposed to CCT action. The cell viability of hepatocytes treated with CCT decreased significantly by 5–12% compared to the control, but was statistically significantly higher by 5–14% compared to doxorubicin, except after CMD-8 and CMT-67 influence, when it does not change. Thus, new local CCT had a selective effect on hepatocytes in vitro and were less hepatotoxic compared to doxorubicin, which may be the basis for further study of its potential in anticancer drugs.

## 1. Introduction

Thiosemicarbazones are a group of sulfur derivatives of semicarbazones that are obtained as a result of the condensation of aldehydes or ketones and thiosemicarbazides in an acidic environment. The structure of thiosemicarbazones has a significant influence on biological activity, and they exhibit pronounced antiviral [1], antidiabetic [2], antifungal, antibacterial and antitumor effects [3]. It should be noted that thiosemicarbazide coordination compounds opened a new era in cancer therapy. Several classes of metal complexes have been synthesized using various forms of ligands and metal ions, and their anticancer activity has been successfully evaluated both in vitro and in vivo [4]. Among them, copper bis(thiosemicarbazone) complexes have attracted more attention because many of them have displayed not only promising anticancer activity [5] but also due to the biological activities of these compounds, which have been shown to be related to their ability to complex metals. Thiosemicarbazone Schiff bases are an important class of compounds in the medicinal and pharmaceutical fields [6].

Most existing antineoplastic drugs can affect normal tissues, often producing cytotoxic effects [7,8]. This fact limits their use and is an indication for a reduction of the dose of the drug, interruption and even complete stopping of the treatment. Despite significant advances in cancer chemotherapy, high systemic toxicity and resistance to antineoplastic chemotherapeutics remain a major challenge for contemporary oncological pharmacotherapy. Many anticancer agents cause severe side effects, which are due to their cytotoxic effects on normal cells.

That is why it is important that anticancer drugs exert high antiproliferative and cytotoxic activity in the tumor cells, without affecting normal tissues. Considering the above concepts, the elaboration and development of new effective chemotherapeutics, with minimal adverse effects, is of great theoretical and practical importance.

In this aspect, of particular interest is the research on the directed synthesis of the coordination compounds of 3D metals carried out at the State University of Moldova in the Advanced Materials in Biopharmaceutics and Engineering Laboratory where, under the leadership of University Professor Academician A. Gulea, a series of new compounds of transition metals with polydentate chelating and macrocyclic ligands were synthesized, and assembled following the condensation of thiosemicarbazide with aldehydes and ketones. The compounds exhibit antitumor properties clearly superior to doxorubicin—a preparation currently widely used in oncology [9,10,11].

Nevertheless, there is no exhaustive research on their influence on the viability of normal liver cells, which are key players in the xenobiotic, including drugs, metabolism. Carrying out such research is very important, because it could contribute to the development of new effective chemotherapies without toxic effects on normal cells.

Various cell viability and cell proliferation assays are used to determine the effect of a test compound on the cells’ state in vitro. The MTT and resazurin test were used for the assay. The physico-chemical properties of MTT (positive charge and lipophilic character of the molecule) ensure its passage through biological membranes into the cell and cellular organelles and its reduction to formazan by the metabolic active cells. Biochemical and histological studies have found formazan in the various intracellular compartments such as plasma membranes, cytoplasm, mitochondria, endoplasmic reticulum, nucleus and microsomes. In addition, some compounds and enzymes (ascorbic acid, tocopherols, dihydrolipoic acid, cysteine, glutathione and glutathione-S-transferase) can also reduce MTT [12,13,14,15]. As a result of the above, the analysis is widely used as a test of the metabolic activity of cells. At the same time, the test was increasingly used to determine secondary processes or cell conditions such as cell viability, cell proliferation, and drug cytotoxicity [12,13,16].

Although the MTT assay is considered to be a golden standard for in vitro cytotoxicity testing and is widely used to test for early cytotoxic events, it is not without limitations. Various factors can cause significant variations in the actual cell viability, including the metabolic activity of the cell, which varies throughout the cell cycle, different culture phases (stationary vs. logarithmic phase), and/or cell type [17], and the presence of reducing compounds such as reduced glutathione, coenzyme A or even the cytotoxic effect of MTT reagent itself that can cause cell damage/apoptosis. In addition, the solubilization step required for the colorimetric quantitation of the formazan product renders this analysis method a lytic point, which prevents further measurements in the sample [18,19].

The resazurin assay is an in vitro cytotoxicity test that allows a fast and inexpensive screening of drug efficacy prior to the in vivo studies. The conversion of resazurin to fluorescent resorufin occurs mostly in the mitochondria, but other enzymes located in the cytoplasm may be able to reduce resazurin [16]. Therefore, the quantity of resorufin generated can be used as an indicator of the metabolic activity of the cell that directly correlates with the number of alive ones [20].

The resazurin method provides precise (high sensitivity and linearity) measurements, does not require any cell lysis, and can be used in a considerable variety of cell models, applying different absorption-based instrumentation approaches. There are ample evidence-based data that support the use of the resazurin assay for cell viability evaluation; nevertheless, many studies have adopted a combined approach for the analysis of cell viability and cytotoxicity in order to obtain the most meaningful and accurate in vitro results [21].

Based on these considerations, the aim of the paper was to study the influence of new copper coordination compounds of thiosemicarbazide on the cell viability and to evaluate the cytotoxic potential in vitro on normal hepatocyte cultures obtained from laboratory animals using the MTT and resazurin tests.

## 2. Materials and Methods

### 2.1. Tested Compounds

Copper coordination compounds, derivatives of thiosemicarbazide (CCT) coded as CMG-41, CMC-34, CMJ-33, CMT-67, TIA-123, CMD-8 and MG-22, were tested in the study (Table 1). The coordinating compounds were synthesized at the Advanced Materials in Biopharmaceutics and Technology Laboratory of the State University of Moldova. Doxorubicin, widely used in cancer treatment, was used as a reference compound.

### 2.2. Hepatocyte Isolation Method

Hepatocytes were isolated from the liver of a group of 3 white male Wistar rats weighing 220–250 g. Before sacrificing the animals, 5000 IU of heparin was injected intraperitoneally. General anesthesia was performed with 60 mg/kg ketamine and 5 mg/kg xylazine, then the fur was removed with a trimmer and processed with 70% alcohol. The thoracoabdominal wall was removed and a suprahepatic portion of the inferior vena cava was cannulated with an 18 G plastic catheter during sustained cardiac contractions to keep the hepatocytes alive.

Next, hepatocytes were isolated by a two-step method [17] using the superior vena cava infusion with 0.05% type II collagenase solution (Himedia, India), 0.01% type I dispase, Ca^2+^ and Mg^2+^ free HBSS (HiMedia, India) with 0.9 mM MgCl_2_, 0.5 mM EDTA and 25 mM HEPES (4-(2-hydroxyethyl)-1-piperazineethane sulfonic acid) (HiMedia, India) [8,9]. The culture medium used was William’s E medium (HiMedia, India) with 2 mM L-glutamine, 5% fetal bovine serum (Lonza, Belgium), antifungal solution with antibiotics (HiMedia, India), 100 nM dexamethasone and 100 nM insulin.

After the enzyme infusion with collagenase and dispase, the liver becomes soft, due to the digestion of the connective tissue. At this point, the liver is transferred to the culture medium and the Glisson capsule is removed to obtain a crude cell suspension which is then filtered, washed and resuspended and diluted to the required cell concentration for culture and testing.

### 2.3. Culture of Hepatocytes and Determination of Hepatocyte Viability with the Trypan Blue Assay

The trypan blue assay is a versatile staining method commonly used to quantify cell death as well as to count cells prior to cell seeding in vitro [22]. This test is based on the ability of the trypan blue molecule, a high molecular weight anionic tetrasulfonate dye, to penetrate the plasma membranes of dead, destroyed cells, which stains intracellular proteins dark blue while viable cells will remain unstained, small and refractive [23]. The number of dead cells can be easily monitored by light microscopic inspection using an automated hemocytometer or cell counter. Cells were counted after treatment with 0.25% trypan blue solution in an automated hemocytometer and cell viability was assessed. (Figure 1).

Liver cells maintain their critical enzyme functionality for a short period after cultivation; therefore, they were used within the first 24 h. For the in vitro studies, the nutrient media was changed every 2 days. When changing the medium in the first 4 h, it was taken into account that the hepatocytes are fragile and can be easily damaged by direct contact; therefore, pipetting was only to be performed from the side of the well. Usually, 1 × 10^8^ cells were extracted from a liver, and the viability of hepatocytes was 88–96%.

### 2.4. Assessment of the Action of CCT on Hepatocyte Viability with the MTT Assay

The viability of the hepatocytes was assessed by the MTT (3-(4,5-dimethylthiazol-2-yl)-2,5-diphenyltetrazolium bromide) colorimetric assay. The MTT test was performed according to the method proposed by Mosmann, with some modifications [24]. The test determines the metabolic activity of cells, based on the reduction of the yellow tetrazolium MTT [3-(4,5-dimethylthiazol-2-yl)-2,5-diphenyl-tetrazolium bromide] by the cellular NAD(P)H-dependent oxidoreductase to insoluble formazan (E,Z)-5-(4,5-dimethylthiazol-2-yl)-1,3-diphenylformazan). The formazan formed can be dissolved with isopropanol, thus obtaining a violet color with an absorption maximum at 570 nm. The intensity of the purple color is directly proportional to the number of living cells, and thus indicates the cell viability.

For the hepatocytes viability evaluation, 100 μL liver cell suspension was poured into a 96-well flat bottom microplate at a density of 1 × 10^4^ cells/well. Cells were allowed to adhere for 24 h at 37 °C in a 3.5% CO_2_ incubator. After 24 h of incubation, the culture medium was replaced with a fresh medium (90 μL). Cells were then treated with 10 µL of test compounds at 10.0 µM/l dilutions and incubated in a CO_2_ incubator for 24 h at 37 °C. Hepatocytes that were not exposed to CCT were used as a control to establish cell viability. Then, 10 µL of MTT working solution (5 mg/mL in phosphate-buffered saline) was added to each well (with and without CCT) and incubated for 4 h at 37 °C until intracellular purple formazan crystals were visible under the microscope. The medium was removed, and the formed formazan crystals were solubilized in isopropanol (100 μL per well) for 30 min at 37 °C. The results were read at 570 nm using Synergy H1 Hydrid Reader multimode microplate spectrophotometer (BioTek Instruments, Winooski, VT, USA).

### 2.5. CCT Cytotoxicity Assessment by Resazurin Assay

For the resazurin cell viability assay, 100 μL liver cell suspension was poured into a 96-well flat bottom microplate at a density of 1 × 10^4^ cells/well. Cells were allowed to adhere for 24 h at 37 °C in a 3.5% CO_2_ incubator. After 24 h of incubation, the culture medium was replaced with a fresh medium (90 μL). Cells were then treated with 10 µL of test compounds at 10.0 µM/L dilutions and incubated in a CO_2_ incubator for 24 h at 37 °C. Hepatocytes that were not exposed to CCT were used as a control to establish the cell viability. After incubation, 20 μL of 0.15 g/L resazurin solution was added to all wells of the microplate. Then, the microplate was incubated for 4 h in the cell culture incubator at 37 °C and 3.5% CO_2_. Absorbance was measured at 570 nm and 600 nm wavelengths using a Power Wave HT microplate spectrophotometer (BioTek Instruments, Winooski, VT, USA).

To assess cytotoxicity, the difference in reduction between the cells treated with the test compounds and the unexposed control cells was calculated, using the formula (1) and expressed in percentages (%).
((O2 × A1) − (O1 × A2)/(O2 × C1) − (O1 × C2)) × 100(1)

In the formula, O1—molar absorption coefficient (ε) of oxidized resazurin at 570 nm; O2 is the molar absorption coefficient (ε) of oxidized resazurin at 600 nm; A1—absorbance of the samples to be investigated at 570 nm; A2—absorbance of the samples to be investigated at 600 nm; C1—absorbance of the positive control at 570 nm; C2—absorbance of the positive control at 600 nm.

### 2.6. Statistical Analysis

All experiments were repeated at least three times. Arithmetic means ± standard error (X ± m) were calculated. To test the significant difference between the studied indices of the compared groups, the post hoc tests for multiple comparisons were applied: Games–Howell after One-Way Anova and Welch’s Anova, and the “*p*” significance threshold (*p* < 0.05) was used. Data were statistically analyzed with the Statistical Package for the Social Science (v.23, SPSS Inc. Chicago, IL, USA).

## 3. Results and Discussions

### MTT Test Results Revealed Low Influence of Hepatocytes’ Viability of the Tested Compounds

The results of the MTT assay showed that the liver cells treated with CCT had a relatively high viability, indicating a low cytotoxicity of the tested compounds. (Figure 2).

Analysis of the results of the MTT test showed that the viability of cells treated with CCT at a concentration of 10.0 μM/L significantly decreased only under the influence of CMC-34 (5.8%, *p* < 0.001) and CMT-67 (12.5%, *p* < 0.01), while other CCTs insignificantly reduced the viability of hepatocytes by 5–11% (*p* > 0.05). The addition of DOXO in a concentration of 10.0 μM/L to the hepatocytes culture induced a non-significant decrease of the hepatocytes’ viability (10%, *p* > 0.05). It can be concluded that CCT at a concentration of 10.0 μM/L did not have a toxic effect on the liver cells, which largely retained their viability, possibly due to preserved metabolic functions.

To determine whether CCTs have a direct effect on cell viability, we tested their effects on the liver cells in vitro using the resazurin test. The results of the evaluation of hepatocytes proliferation under the influence of thiosemicarbazide coordination compounds at a dose of 10.0 μM/L, assessed using the resazurin test, are shown in Figure 3

The resazurin test results revealed that the proliferation during the incubation period of the control culture of the hepatocytes was 12.99%.

The effect of CCT 10.0 μM/L on the proliferation of hepatocyte culture, assessed using the resazurin test, differed from the effect of CCT at the same dose, established on viability using the MTT test. Thus, the statistically significant modifications of the hepatocytes proliferation were attested when the hepatocytes were treated with CMG-41 (+2%, *p* < 0.01) and CMC-34 (−17.7%, *p* < 0.01), the statistically non-significant increase was produced by MG-22 (22.51%, *p* > 0.05) and DOXO (27.69%, *p* > 0.05), while there was a statistically non-significant decrease by CMJ-23 (17.34%, *p* > 0.05), TIA-123 (17.05%, *p* > 0.05), CMD-8 (101.32%, *p* > 0.05) and CMT-67 (7.73%, *p* > 0.05).

It can be concluded that most of the seven CCTs tested at a dose of 10 μM/L do not significantly affect either the viability or the proliferation of liver cell culture in vitro. The results obtained reveal the individual response of hepatocyte culture cells (direction and magnitude) to the action of various CCTs. This can be a demonstration of the ability of the studied CCTs to form adaptive responses of liver cells to the action of these compounds, through which liver cells improve their functionality, a process known as hormesis [25]. In medicine, hormesis is defined as the adaptive response of cells to moderate and usually intermittent stress such as exercise, caloric restriction, exposure to low doses of certain phytochemicals and ischemic preconditioning. Hormetic responses are mediated by cellular signaling pathways and molecular mechanisms that include enzymes (kinases and deacetylases) and transcription factors (Nrf-2 and NF-kappaB) and lead to an increase in the production of protective and regenerative compounds in the cell (growth factors, phase 2 enzymes, antioxidants, chaperones, etc.) [26].

The tested CCTs contain copper and the hormetic response can be determined by the metal, which was administrated. Many life-sustaining biological reactions involve transition metals, which usually coordinate to the O- or N-terminus of proteins in various ways and play a fundamental role in ensuring the conformation and function of biological macromolecules [27]. Copper is an essential component of the prosthetic group of cytochrome oxidase, catalase, superoxide dismutase, alcohol dehydrogenase, lysyl oxidase, and alkaline phosphatase [28], which are involved in ATP synthesis, antioxidant protection, synthesis of the components of the extracellular matrix, etc. [29].

It has been shown that copper-containing coordination compounds are promising antitumor therapeutic agents acting through various mechanisms, such as the inhibition of proteasome activity [30,31,32], the formation of reactive oxygen species (ROS) [33,34], DNA degradation [35], DNA intercalation [36], paraptosis [37], and others. Most Cu(II) coordination compounds rapidly form adducts with glutathione in the cell, resulting in the formation of a univalent Cu(I) coordination compound capable of generating superoxide anion, which can induce the formation of ROS in the Fenton reaction [38] and be responsible for the cellular effects of the copper-containing coordination compounds in general and the CCTs tested in our study in particular.

Malignant and inflamed tissues metabolize an increased amount of copper compared to healthy tissues [39], which gives copper-containing coordination compounds an additional advantage over other metal-containing drugs that can be used in the cancer treatment. The development of new copper coordination compounds with antitumor activity is a promising and relevant area of medical chemistry [40,41,42], which is confirmed by the results of our studies on the effect of CCTs on liver cell viability and proliferation.

## 4. Conclusions

Seven coordinating compounds of thiosemicarbazide (CMC-34, CMJ-33, CMG-41, CMT-67, TIA-123, CMD-8 and MG-22) were studied in order to select the least cytotoxic compounds that maintain the state of the hepatocytes at the level of the control cell culture. The results of the study show that the response of liver cells in an in vitro experiment to the action of coordinating compounds of thiosemicarbazide is selective. The demonstrated selectivity may underlie their individual cytotoxic and antiproliferative effects on the cells. The data obtained indicate the need to continue research of the effect of CCT on cells in order to establish the mechanisms of the revealed effects, as well as the features of the effect on other types of cells, including cancer cells.

## Figures and Tables

**Figure 1 biomedicines-11-00366-f001:**
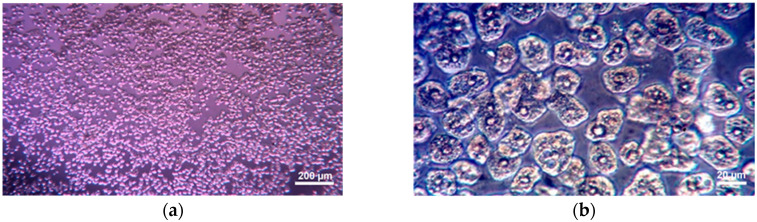
Liver cells visualized with the inverted microscope with KZD phase contrast. (**a**)—10×, (**b**)—40×.

**Figure 2 biomedicines-11-00366-f002:**
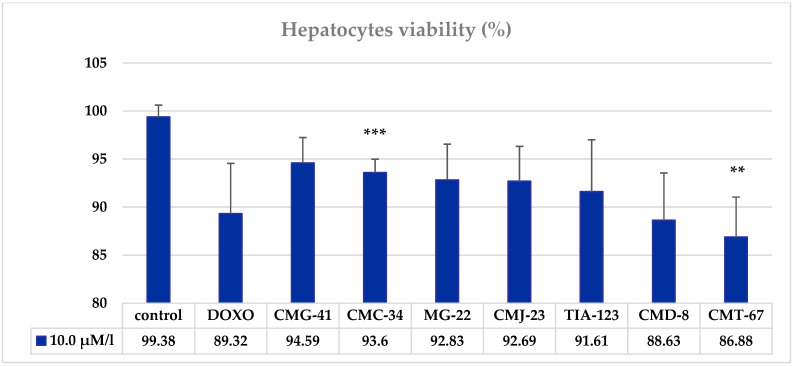
Influence of thiosemicarbazide coordination compounds at doses of 10.0 μM/L on Wistar rats’ hepatocytes’ viability tested in vitro by MTT assay. Note: statistical significance in comparison with the control—**—*p* < 0.01; ***—*p* < 0.001.

**Figure 3 biomedicines-11-00366-f003:**
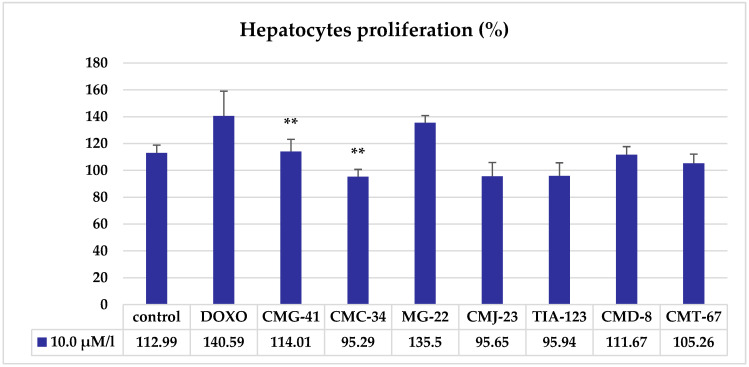
Proliferation of hepatocytes under the influence of thiosemicarbazide coordination compounds at a dose of 10.0 μM/L assessed by the resazurin test. Note: statistical significance in comparison with the control—**—*p* < 0.01.

**Table 1 biomedicines-11-00366-t001:** The coordination compounds of thiosemicarbazide tested in the study.

Code	Chemical Name of the Substance
CMC-34	Chloro-{N’-[phenyl(pyridin-2-yl)methylidene]-N-pyridin-2-ylcarbamohydrazonothioato}copper
CMJ-33	Chloro-{4-(3 methoxy phenyl)-2-[1-(pyridin-2-yl)ethylidene] hydrazine-1-carbothioamido} copper
CMG-41	Nitrato-{N’-[phenyl(pyridin-2-yl)methylidene]-N-prop-2-en-1-ylcarbamohydrazonothiato} copper
CMT-67	Nitrato-{N-phenyl-N’-(pyridin-2-ylmethylidene)carbamohydrazonothioato} copper
TIA-123	Chloro-{N’-[phenyl(pyridin-2-yl)methylidene]-N-prop-2-en-1-ylcarbamohydrazonothioato}copper
CMD-8	Chloro-{4-ethyl-2-[phenyl (pyridin-2-yl)methylidene] hydrazine-1-carbothioamido} copper
MG-22	Chloro-{N’-(4-methoxyphenyl)-N,N-dimethylcarbamimidothioato}copper
DOXO	Doxorubicin

## Data Availability

The datasets are available from the corresponding authors upon reasonable request.

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
