# Peer review of "In Vitro Evaluation of the Cytotoxic Potential of Thiosemicarbazide Coordinating Compounds in Hepatocyte Cell Culture"

_biomedicines, 2023, doi:10.3390/biomedicines11020366_

Round 1

Reviewer 1 Report

The manuscript reported the cell viability and cytotoxicity of some materials of coordination compounds of thiosemicarbazide. The manuscript's novelty and its applications are not clear. Besides the English language needs more polishing.

Please see the following comments:

1-         The abstract section did not show the work's novelty. It is highly recommended to redesign the abstract section.

2-         The English language needs more polishing; it is highly recommended to revise the manuscript language. It is better to check these sentences and others “Doxorubicin at concentrations of 10.0 μM/L was used as the reference substance.”; “Cytotoxicity and cell viability 18 of hepatocytes was determined in vitro with the aim of evaluating the action of some coordination 19 compounds of thiosemicarbazide (CCT), CMC-34, CMJ-33, CMG-41, TIA-123, MG-22, CMT- 67 and 20 CMD-8.”

3-         In Figure 2, The authors already wrote down the full name of the thiosemicarbazide complexes, so no need to rewrite the full names. In addition, it is highly recommended to put the error bare for each column.

4-         From Figure 2, The % viability values were significantly decreased under the treatment by CMD-8, and CMT not for CMC and CMT as the authors described. Please check this part carefully.

5-         Why the authors added the p-values for CMC and CMD only? What about the other materials?

6-         I recommend moving this part into the introduction “The physico-chemical properties of MTT (positive charge and lipophilic character of the molecule) ensure its passage through biological membranes into the cell and cellular organelles and its reduction to formazan by the active metabolic cells. Biochemical and histological studies have found formazan in various intracellular compartments such as plasma membranes, cytoplasm, mitochondria, endoplasmic reticulum, nucleus and microsomes. In addition, some compounds and enzymes (ascorbic acid, tocopherols, dihydrolipoic acid, cysteine, glutathione and glutathione-S-transferase) can also reduce MTT 202 [16-19]. As a result of the above, the analysis is widely used as a test of the metabolic activity of cells. At the same time, the test increasingly was used to determine secondary processes or cell conditions such as cell viability, cell proliferation, and drug cytotoxicity [16, 17, 20]. Nitrato-{N'-[phenyl(pyridin-2-yl)methylidene]-N-prop-2-en-1-ylcarbamohydrazono Although the MTT assay is considered the gold standard for in vitro cytotoxicity testing and is widely used to test for early cytotoxic events, it is not without limitations. Various factors can cause significant variations in actual cell viability, including the metabolic activity of the cell, which varies throughout the cell cycle, different culture phases (stationary vs. logarithmic phase), and/or cell type [12], the presence of reducing compounds such as reduced glutathione, coenzyme A or even the cytotoxic effect of MTT reagent itself, that can cause cell damage/apoptosis. In addition, the solubilisation step required for the colorimetric quantitation of the formazan product renders this analysis method a lytic point, preventing further measurements in the sample [21, 22].”

7-         It is highly recommended to add the fluorescence microscope image of the cells dying.

8-         The live-dead assay is one of the techniques that provide the author’s work. It is highly recommended to use this technique.

Author Response

Response to Reviewer 1 Comments

1-         The abstract section did not show the work's novelty. It is highly recommended to redesign the abstract section.

The abstract section was redesigned.

2-         The English language needs more polishing; it is highly recommended to revise the manuscript language. It is better to check these sentences and others “Doxorubicin at concentrations of 10.0 μM/L was used as the reference substance.”; “Cytotoxicity and cell viability 18 of hepatocytes was determined in vitro with the aim of evaluating the action of some coordination 19 compounds of thiosemicarbazide (CCT), CMC-34, CMJ-33, CMG-41, TIA-123, MG-22, CMT- 67 and 20 CMD-8.”

The article was edited grammatically and stylistically.

3-         In Figure 2, The authors already wrote down the full name of the thiosemicarbazide complexes, so no need to rewrite the full names. In addition, it is highly recommended to put.

The full names of the CCT were removed.

The error bare for each column was added in figures 2 and 3.

4-         From Figure 2, The % viability values were significantly decreased under the treatment by CMD-8, and CMT not for CMC and CMT as the authors described. Please check this part carefully.

The description in the text was according to the data presented in the figure nr. 2 and the results of the statistical evaluation of the raw data. Just in case of CMC-34 (5.8%, p<0.001) and CMT-67 (12.5%, p<0.01) the differences of the experiment compared with the control were statistically significant, while in case of CMD-8 and other CCTs the decrease was not statistically significant (p>0.05).

5-         Why the authors added the p-values for CMC and CMD only? What about the other materials?

The "p" values were presented only for statistically significant differences in order not to overload the figures with data.

6-         I recommend moving this part into the introduction “The physico-chemical properties of MTT (positive charge and lipophilic character of the molecule) ensure its passage through biological membranes into the cell and cellular organelles and its reduction to formazan by the active metabolic cells. Biochemical and histological studies have found formazan in various intracellular compartments such as plasma membranes, cytoplasm, mitochondria, endoplasmic reticulum, nucleus and microsomes. In addition, some compounds and enzymes (ascorbic acid, tocopherols, dihydrolipoic acid, cysteine, glutathione and glutathione-S-transferase) can also reduce MTT 202 [16-19]. As a result of the above, the analysis is widely used as a test of the metabolic activity of cells. At the same time, the test increasingly was used to determine secondary processes or cell conditions such as cell viability, cell proliferation, and drug cytotoxicity [16, 17, 20]. Although the MTT assay is considered the gold standard for in vitro cytotoxicity testing and is widely used to test for early cytotoxic events, it is not without limitations. Various factors can cause significant variations in actual cell viability, including the metabolic activity of the cell, which varies throughout the cell cycle, different culture phases (stationary vs. logarithmic phase), and/or cell type [12], the presence of reducing compounds such as reduced glutathione, coenzyme A or even the cytotoxic effect of MTT reagent itself, that can cause cell damage/apoptosis. In addition, the solubilisation step required for the colorimetric quantitation of the formazan product renders this analysis method a lytic point, preventing further measurements in the sample [21, 22].”

The fragment has been moved to the Introduction section. As a result, in order to ensure a logical presentation of the material in the section, minor adjustments and additions have been made to the Introduction section.

7-         It is highly recommended to add the fluorescence microscope image of the cells dying.

We did not take fluorescent images of the cells, because the  BioTek Synergy H1 modular multimode microplate reader (USA) is not providing microscope functions, but can be used for the measurement of absorbance, luminescence, UV-Vis etc

8-         The live-dead assay is one of the techniques that provide the author’s work. It is highly recommended to use this technique.

The authors are grateful for the reviewer's recommendation to use the live-dead test in our research and will expand its application in our work.

Reviewer 2 Report

The manuscript describes experimental work which seems to have been executed to a good standard and is both novel and important. However, I have major worries about the statistical analysis which defies comprehension. For example in Fig. 3 the compound CMD-8 has the largest effect (seems to cause 90% inhibition) and yet its effect is reported as being of no statistical significance. Also, CMG-41 has neglegible effect as an inhibitor (indeed it causes 2% activation) and yet it is reported as having p<0.01 i.e. highly significant. This referee is not a specialist in these assays so it is not clear how this sort of disparity can arise but maybe it is to do with the scatter in the measurements and if so then this should be explained in the manuscript. However, I fear that the statistics must be very thoroughly checked before publication should proceed. 

Author Response

Response to Reviewer 2 Comments

The manuscript describes experimental work which seems to have been executed to a good standard and is both novel and important. However, I have major worries about the statistical analysis which defies comprehension. For example in Fig. 3 the compound CMD-8 has the largest effect (seems to cause 90% inhibition) and yet its effect is reported as being of no statistical significance. Also, CMG-41 has neglegible effect as an inhibitor (indeed it causes 2% activation) and yet it is reported as having p<0.01 i.e. highly significant. This referee is not a specialist in these assays so it is not clear how this sort of disparity can arise but maybe it is to do with the scatter in the measurements and if so then this should be explained in the manuscript. However, I fear that the statistics must be very thoroughly checked before publication should proceed. 

As it was mentioned in the section 2.6. Statistical analysis, the significant difference between the studied indices of the compared groups were evaluated based on the post hoc tests for multiple comparisons: Games-Howell after One-Way Anova and Welch's Anova, using the Statistical Package for the Social Science (v.23, SPSS Inc. Chicago, IL, USA) that were suitable for the raw data that were obtained in the study. Anova statistical methods were applied because we needed to compare multiple groups (9 groups – 1 control, 7 CCT and 1 doxorubicin). The Games-Howell post hoc test was applied to compare all possible combinations of group differences because the groups were not homogenous (normality, equal variances or sample sizes).

Round 2

Reviewer 1 Report

Even though I rejected the paper in the first revision, the authors have carefully revised and modified the manuscript. therefore, I recommend accepting the manuscript in its current form. 

Reviewer 2 Report

I am not able to judge the analysis of the results so I must defer to the other referee's judgement on that. It seems that the results for certain compounds are coming out as statistically significant if their variance is low rather then there being a visible difference from the control in the graphs.